# Attenuation of the BOLD fMRI Signal and Changes in Functional Connectivity Affecting the Whole Brain in Presence of Brain Metastasis

**DOI:** 10.3390/cancers16112010

**Published:** 2024-05-25

**Authors:** Pia Angstwurm, Katharina Hense, Katharina Rosengarth, Quirin Strotzer, Nils Ole Schmidt, Elisabeth Bumes, Peter Hau, Tobias Pukrop, Christina Wendl

**Affiliations:** 1Faculty of Medicine, University of Regensburg, 93053 Regensburg, Germany; 2Center for Neuroradiology, Institute for Diagnostic Radiology, University Hospital Regensburg, 93053 Regensburg, Germany; quirin.strotzer@klinik.uni-regensburg.de (Q.S.); christina.wendl@klinik.uni-regensburg.de (C.W.); 3Department of Neurosurgery, University Hospital Regensburg, 93053 Regensburg, Germany; katharina.hense@klinik.uni-regensburg.de (K.H.); katharina.rosengarth@klinik.uni-regensburg.de (K.R.); nils-ole.schmidt@klinik.uni-regensburg.de (N.O.S.); 4Department of Neurology, University Hospital Regensburg, 93053 Regensburg, Germany; elisabeth.bumes@klinik.uni-regensburg.de (E.B.); peter.hau@klinik.uni-regensburg.de (P.H.); 5Department of Haematology and Internal Oncology, University Hospital Regensburg, 93053 Regensburg, Germany; tobias.pukrop@klinik.uni-regensburg.de

**Keywords:** brain metastasis, functional magnetic resonance imaging, blood oxygenation level-dependent imaging, functional connectivity, percent signal change

## Abstract

**Simple Summary:**

Functional MRI has become established in the surgery of brain metastases (BM) as a preoperative diagnostic tool to identify intact eloquent cortex areas. Evidence shows that by allowing surgeons to spare these intraoperatively, patients’ postoperative outcome is improved in terms of significantly reduced mortality and morbidity. The influence specifically of BM on the fMRI signal and brain networks has scarcely been investigated, as most studies to date refer only to primary brain tumors or include various tumor entities. Our work examined how BM affect cortical activation and brain networks using task-based fMRI. We found a qualitative attenuation of patients’ fMRI signal in the metastasis-affected hemisphere compared to the contralateral hemisphere and alterations in all examined brain networks of the patients compared to healthy controls, and also in the contralateral hemisphere. Thus, our results provide insights into the behavior of BM during fMRI examination and their impact on the integrity of the brain.

**Abstract:**

To date, there are almost no investigations addressing functional connectivity (FC) in patients with brain metastases (BM). In this retrospective study, we investigate the influence of BM on hemodynamic brain signals derived from functional magnetic resonance imaging (fMRI) and FC. Motor-fMRI data of 29 patients with BM and 29 matched healthy controls were analyzed to assess percent signal changes (PSC) in the ROIs motor cortex, premotor cortex, and supplementary motor cortex and FC in the sensorimotor, default mode, and salience networks using Statistical Parametric Mapping (SPM12) and marsbar and CONN toolboxes. In the PSC analysis, an attenuation of the BOLD signal in the metastases-affected hemisphere compared to the contralateral hemisphere was significant only in the supplementary motor cortex during hand movement. In the FC analysis, we found alterations in patients’ FC compared to controls in all examined networks, also in the hemisphere contralateral to the metastasis. This indicates a qualitative attenuation of the BOLD signal in the affected hemisphere and also that FC is altered by the presence of BM, similarly to what is known for primary brain tumors. This transformation is not only visible in the infiltrated hemisphere, but also in the contralateral one, suggesting an influence of BM beyond local damage.

## 1. Introduction

Brain metastases (BM) are a severe and frequent complication of systemic cancer [1] and the most common type of intracerebral neoplasms [2]. Various primary tumors are able to metastasize to the brain, with lung cancer, breast cancer, and malignant melanomas being the most frequent ones [2]. Even though survival of affected patients has improved in recent years due to earlier detection and innovative therapies, patients with an untreated BM still show a median survival of only a few months [3,4]. Additionally, BM frequently cause grave neurological symptoms and considerably reduced quality of life [5].

In surgery of brain metastases, functional magnetic resonance imaging (fMRI) has become established in the preoperative workup to identify eloquent functional cortical areas and increase the rate of maximal safe resections. By enabling surgeons to resect as much tumor tissue as possible and spare eloquent areas, the postoperative outcome of patients with brain tumors can be improved in terms of a significantly reduced mortality and morbidity [6]. Nevertheless, there are hardly any studies that specifically investigate the influence of BM on the fMRI signals. Using fMRI to examine patients with brain metastases has both advantages and disadvantages compared to other imaging techniques such as CT, PET, and SPECT. The most important advantage of fMRI is the ability to dynamically detect the functionality and activity of apparently healthy brain areas as well as functional connectivity while patients perform tasks. The other imaging techniques, however, are used to localize and classify tumors. In this way, fMRI is useful to identify intact eloquent cortex areas preoperatively so that surgeons can specifically preserve them intraoperatively. This can significantly reduce mortality and morbidity of patients with brain tumors [6]. Disadvantages of fMRI compared to other imaging techniques are the burden to which patients are exposed when performing the tasks. In addition, specially trained personnel are required to analyze and interpret these data.

However, a number of studies have already shown an alteration of the fMRI BOLD (blood oxygenation level dependent) signal in patients with brain tumors in general, in terms of attenuated activation in the affected hemisphere compared to the contralateral one [7,8,9,10,11,12,13]. In order to identify intact functional areas, the fMRI BOLD signal uses neurovascular coupling, which may be disturbed especially in the vicinity of higher-grade gliomas due to pathologic neovascularization [10]. However, other factors such as altered concentrations of metabolic factors and neurotransmitters, as well as a disturbed relationship between neurons, astrocytes, and capillaries, may influence neurovascular coupling and thus the fMRI BOLD signal in the presence of brain tumors [10,14]. Therefore, the reliability of fMRI in the immediate surroundings of these tumors must always be critically questioned prior to resection of possibly intact regions.

Most published studies are restricted to primary brain tumors, while only occasionally metastatic patients were also included [15]. Hua et al. showed a change in FC in metastatic patients towards a more random organization of networks compared with healthy controls, and even suggested the possibility of an increase in these changes in FC due to surgical resection of the metastasis [16]. It was found that primary brain tumors are able to cause alterations and a decrease in FC that are not limited to the tumor area, but also affect the contralateral hemisphere [17]. Compared to healthy subjects, FC in brain tumor patients is severely changed. In an fMRI study by Maesawa and colleagues, FC was found to be partly enhanced and partly attenuated in patients with gliomas compared to controls, an effect that was also measurable in the hemisphere contralateral to the tumor [18]. The changes in FC here correlated in part with cognitive functions [18]. A study using fMRI in patients with gliomas demonstrated decreased FC in patients compared to controls [19]. This decrease was more pronounced in high grade gliomas than in low grade gliomas and correlated with motor deficits in low grade gliomas, while tumor volume and distance to the motor cortex had no effect [19]. Ghumman and colleagues found attenuated default mode network (DMN) connectivity in patients with left hemispheric tumors using fMRI in a sample of patients with different tumor entities [15]. A study using fMRI in patients with gliomas found abnormal connectivity detectable even in regions distant from the tumor, which was associated with tumor biology and cognitive function [20]. Tordjman and colleagues demonstrated partly attenuated, partly enhanced FC using fMRI in patients with gliomas, but found no influence of WHO grade and tumor biology [21].

Knowing that pathophysiology of BM is significantly different to high grade gliomas, the aim of this study was to examine whether findings in primary brain tumors can also be applied to brain metastases. We hypothesized, in agreement with previous study results, that activation of the fMRI BOLD signal in the affected hemisphere compared to the contralateral one would be attenuated. We also speculated that BM alter and disrupt functional connectivity in a manner similar to primary brain tumors, and that this impact is not limited to the immediate area surrounding the metastasis.

## 2. Materials and Methods

### 2.1. Study Sample

Our study was designed and carried out in accordance with the guidelines of the Declaration of Helsinki and was positively voted by the ethics committee of the University Hospital Regensburg (protocol code: 22-2934-104, 18 May 2022).

The study sample included patients with a confirmed diagnosis of at least one brain metastasis, who obtained fMRI during routine clinical diagnostics at the University Hospital Regensburg between 2015 and 2022. During fMRI examination, patients performed movements of the separate hands and feet, respectively.

In total, 29 patients were selected (mean age 59.90 years; range 31–85 years). Table 1 shows an overview of the sample sizes, sociodemographic, and clinical data of the patients. Since both hands and feet were not examined for all patients, a distinction is made here between patients having performed hand movements and foot movements as a paradigm. Previous therapies were assessed retrospectively. After a thorough review of the medical charts, it remained unclear for three patients whether they had already received radiotherapy or system therapy prior to the fMRI.

Preoperative neurological deficits were also collected in a retrospective manner. Fifteen of the patients had persistent sensorimotor deficits that were temporally related to the occurrence of the metastasis. Three additional patients had a transient sensorimotor deficit that was no longer present at the time of neurological examination. Seventeen patients had various other neurological deficits, e.g., impaired memory or concentration. Only two patients were asymptomatic.

Patients were retrospectively assigned to two subgroups depending on whether the metastasis was located in the left or right hemisphere. The division into these two subgroups aimed at a more specific identification of the brain areas activated by the respective motor paradigms. In twelve cases, metastases were present in both hemispheres. Whether these were assigned to the left or right hemispheric group was decided according to which hemisphere was stronger affected, meaning that either the largest lesion or a greater number of lesions were located there. The group of patients with left hemispheric and the group of patients with right hemispheric metastases were tested for a difference in metastatic volume to exclude an inter-group bias. The test did not yield a significant result (t(24) = −1.077; *p* = 0.292).

MRI data of the control group were selected from the publicly available Human Connectome Project database [22]. From these, sex-matched control subjects were selected while age-matching was not possible, as the oldest subject in the database is 37 years old (mean age 35.24 years; range 31–37 years). Thus, the mean age and standard deviations differed significantly (t(28.410) = 11.464; *p* < 0.001) between the two groups.

### 2.2. Image Acquisition

Patients were examined at the University Hospital Regensburg with a Siemens 3 Tesla whole-body scanner (MAGNETOM Skyra, Siemens, Erlangen, Germany) with a 32-channel head coil. Before the measurement, the examination procedure was explained to the patients in detail and practiced with them in order to ensure they could perform the motor tasks independently. We used a MR-compatible 32-inch BOLD screen (Cambridge Research Systems, Rochester, UK), with which cue stimuli for the tasks were presented to the patients. This screen could be seen by the patients via a mirror attached to the head coil. The fMRI images were obtained using a T2-weighted echo planar imaging (EPI) sequence (TR = 3000 ms, TE = 35 ms, flip angle = 90°, FoV = 1151 × 1151 mm^2^, voxel size = 2 × 2 × 3 mm^3^, 28 slices). In addition to the functional images, a high-resolution T1-weighted 3D data set (TR = 2300 ms, TE = 2910 ms, flip angle = 9°, FoV = 256 × 256 mm^2^, voxel size = 1 × 1 × 1 mm^3^) was generated for structural images.

Control group data were provided by the Human Connectome Project, WU-Minn Consortium (Principal Investigators: David Van Essen and Kamil Ugurbil; 1U54MH091657) funded by the 16 NIH Institutes and Centers that support the NIH Blueprint for Neuroscience Research, and by the McDonnell Center for Systems Neuroscience at Washington University. Data were collected between 2012 and 2015 at Washington University in St. Louis (MO, USA), on a modified 3 Tesla Skyra scanner with a 32-channel head coil [22]. Cue stimuli for the tasks were projected via a beamer onto a screen that subjects could see via a mirror attached to the head coil. Functional images were created using an EPI sequence (TR = 720 ms, TE = 33.1 ms, flip angle = 52°, FoV = 208 × 180 mm^2^, voxel size = 2 × 2 × 2 mm^3^, 72 slices) [23,24]. For T1-weighted images, TR was 2400 ms, TE was 2.14 ms, flip angle was 8°, FoV was 224 × 224 mm^2^, and voxel size was 0.7 × 0.7 × 0.7 mm^3^ [25].

### 2.3. Paradigms

During fMRI examination, patients and controls had to perform movements of the hands and feet. The acquisition paradigms differed between both groups. In the patient group, a separate run was conducted for each extremity studied (e.g., left hand). Here, only the regions affected by the metastasis were examined, depending on the localization of the metastasis. The regions that were not affected by the metastasis were excluded from the examination. This is why in some patients, only the hands or only the feet were studied, but always the respective extremity of both sides to allow for the comparability between the affected and contralateral hemispheres. Thus, as required, between two and four runs per patient took place, one each for the left hand, right hand, left foot, or right foot. The paradigm started with baseline condition followed by a block of task condition. Each block lasted 24 s and was repeated four times for each body part studied. During baseline, the screen was red, signaling to the patients to lie still, whereas during the task, the screen was green, indicating to the patients to perform the respective movement. The patients’ motor task for the hands was to touch the tip of the thumb of each hand with one of the other four fingers in turn, and the task for the feet was a grasping motion of the toes.

Within the Human Connectome Project, both hands, both feet, and the tongue were studied for each healthy control. For patients, however, we focused only on the movement of the hands and feet, since only one hemisphere is addressed at a time, thus ensuring good differentiation and comparability between the two hemispheres. The fMRI investigation of the controls consisted of two runs, one with right-to-left and the other with left-to-right phase encoding [23]. Since a control subject was explicitly assigned to a specific patient, movement conditions not examined in the individual patient were also excluded here. The motor task differed slightly compared to the patient group. The controls’ task for the hands was a lifting of the thumb, index finger, and little finger, one at a time. Their task for the feet was an alternation of plantar flexion and dorsal extension, and the task for the tongue was a touching of the right, middle, and left parts of the upper row of their teeth [26].

### 2.4. Data Analysis

Statistical Parametric Mapping 12 (SPM12; http://www.fil.ion.ucl.ac.uk/spm (accessed on 24 May 2024)) implemented in Matlab 2020a (The Mathworks Inc., Natick, MA, USA) was used to analyze the fMRI images. Before statistical analysis, images were preprocessed in the same way for the patient and control subject data. Realignment, co-registration to the T1-weighted image, segmentation, spatial normalization to the standard Montreal Neurological Institute (MNI) space, and spatial smoothing of the images were performed. A three-dimensional Gaussian function (FWHM 8 mm) was used for this purpose. After preprocessing, a first level analysis based on the general linear model (GLM) [27] was carried out. Therefore, an individual design matrix was created from the regressors representing the motor task performed by the subjects. The head movement parameters measured in the first step of preprocessing were also used as regressor. These regressors were then convolved with a model of the hemodynamic response function.

Regions of interest (ROI) analyses were performed using the Marsbar toolbox in SPM [28]. The anatomical ROIs motor cortex (BA4), premotor cortex (BA6), and supplementary motor cortex were first exported from the Anatomy Toolbox Version 3.0 [29,30,31]. Then, the global maximum activation and its coordinates in the ICBM152 [32,33] standard brain were determined for each region. To create the ROIs, these coordinates were set as the center of a sphere with a radius of 5 mm. Thus, for each subject, six ROIs were created (three ROIs per hemisphere). The rationale behind this approach was that the size of an ROI is a significant factor influencing the PSC, as larger ROIs contain more voxels, which may have different signal characteristics and noise levels, resulting in higher variability within the ROI, which in turn may affect the mean PSC. ROIs that overlapped with one of the metastases were excluded to avoid biasing the results. For each ROI, percent signal change (PSC) between resting and activation states during the motor task was calculated.

The CONN toolbox version 22.a [34], which is implemented in SPM, was used to analyze functional connectivity (FC). Since the preprocessing is identical to the PSC analysis, the preprocessed images of the patients and controls were imported. Then, de-noising was performed using the head motion parameters measured at the beginning of preprocessing. Functional data were filtered with a bandpass filter (0.008–0.09 Hz) to reduce noise artifacts [35]. First and second level seed-to-voxel analyses were performed. For this purpose, 14 seeds were selected from the sensorimotor network (SMN), as well as the default mode network (DMN) and the salience network (SN), so as to investigate a global influence of the metastasis. During the first-level analysis, we calculated Fisher-transformed bivariate correlation between the mean BOLD time series of the seed ROI and the BOLD time series of each voxel of the brain, resulting in seed-based connectivity maps for each subject. The patients, as well as their assigned control subjects, were subdivided according to the hemisphere in which the largest metastasis of the patients was located in order to prevent the results from being distorted by hemispherical effects. Subsequently, the Fisher-transformed bivariate correlation coefficients were compared in the second-level analysis, and significant differences between the groups were identified.

### 2.5. Statistical Analysis

Statistical tests of numerical data, such as percent signal change, were performed using SPSS 26 Statistics software (IBM, SPSS Statistics, Armonk, NY, USA). In the case of violations of the sphericity assumption, the Greenhaus–Geisser correction was used. Adjustment for multiple testing was performed using the Benjamini–Hochberg correction (False Discovery Rate; FDR) [36]. The respective standard errors of the mean are illustrated by the error bars. Statistically significant results are marked with * for *p* < 0.05.

## 3. Results

### 3.1. Percent Signal Change

The percent signal change of the patients was calculated for the ROIs motor cortex (BA4), premotor cortex (BA6), and supplementary motor cortex (SMA) in the left and right hemisphere, respectively, and then compared between affected and unaffected hemisphere (see Figure 1). Here, we found a significant difference between the two hemispheres in the SMA during movement of the hands (t(27) = −3.008; p_uncorr._ = 0.006; p_adj._ = 0.018; d_z_ = 0.568). In addition to that, there was a trend for the BA4 during the hand movements, which was no longer existent after correction for multiple testing (t(27) = −1.874; p_uncorr._ = 0.072; p_adj._ = 0.108; d_z_ = 0.354). The other ROIs showed no significant differences, neither for the paradigms of the hands (BA6: t(27) = −1.649; p_uncorr._ = 0.111; p_adj._ = 0.111; d_z_ = 0.312), nor for the paradigms of the feet (BA4: t(15) = −0.902; p_uncorr._ = 0.381; p_adj._ = 0.381; d_z_ = 0.225; BA6: t(15) = −1.195; p_uncorr._ = 0.251; p_adj._ = 0.377; d_z_ = 0.299; SMA: t(15) = −1.367; p_uncorr._ = 0.192; p_adj._ = 0.377; d_z_ = 0.342).

### 3.2. Functional Connectivity

Functional connectivity (FC) of patients and healthy control subjects was investigated using seed-to-voxel analysis for the networks sensorimotor network (SMN), default mode network (DMN), and salience network (SN). A graphical representation of the differences in the FC of both groups is given in Figure 2, Figure 3, Figure 4, Figure 5, Figure 6, Figure 7, Figure 8 and Figure 9. The respective clusters with MNI coordinates, sizes, *p*-values, and regions are listed in Appendix A.

Figure 2, Figure 3, Figure 4, Figure 5, Figure 6, Figure 7, Figure 8 and Figure 9 show for each seed of the three investigated networks the significant differences in the seed-to-voxel connectivity to other brain areas. Areas shown in red to yellow are more strongly connected to the examined seeds in controls, while areas shown in blue to purple are more strongly connected to the seeds in patients. We found significant differences between patients and healthy controls for all three networks studied. These differences were not only observed in the hemisphere affected by the metastasis, but also in the contralateral hemisphere. As the Figures demonstrate, FC was mainly attenuated in patients compared to controls. However, there were also some connections that were more pronounced in patients. Particularly the superior seed of the SMN and the seeds in the left and right rostral prefrontal cortices of the SN showed such connections.

## 4. Discussion

This retrospective study aimed to examine how the presence of BM affects cortical activation and functional connectivity. To investigate this research question, we used motor paradigms in task-based fMRI to assess and analyze cortical activation via PSC as well as FC in patients with brain metastases.

The analysis of the patients’ PSCs showed a significant difference between affected and unaffected hemispheres only for the SMA during hand movements, as well as a trend for the motor cortex (BA4) in the paradigms of the hands which was no longer existent after correction for multiple testing. Although the difference was not significant for the other regions, they also showed a qualitative attenuation of the BOLD signal in the metastasis-affected hemisphere compared to the contralateral hemisphere. We found high standard errors of the mean for the motor cortex and premotor cortex during movements of the hands. The high variance of the data may be responsible for the fact that the difference between the two hemispheres was not significant for these two regions. This aligns with published data showing an attenuation of the BOLD signal in the affected hemisphere compared to the contralateral hemisphere for brain tumors in general [7,8,9,10,11,12,13].

In FC analysis, significant differences between patients and healthy controls were found for all three networks studied, not only in the affected hemisphere, but also in the contralateral hemisphere. FC was mainly attenuated in patients compared to controls. Particularly the superior seed of the SMN and the seeds in the left and right rostral prefrontal cortices of the SN also showed connections that were more pronounced in patients. Previous studies on FC in primary brain tumors, respectively, showed that FC was severely altered by the influence of a brain tumor, not only in the immediate surroundings of the lesion, but also in more distant areas [17,18,20].

The results of our study suggest that BM has an effect on fMRI signal and functional connectivity that is comparable to that of primary brain tumors [11,18]. The fMRI signal in the affected hemisphere tended to be attenuated compared to the contralateral hemisphere. FC is diffusely disrupted in the presence of BM. It was mostly attenuated in patients, but there were also a few connections that were stronger in patients than in controls. This impressively supports the hypothesis of Hua et al. that FC is more randomly organized in the presence of BM than in healthy subjects, thus causing a less efficient and properly operating interaction between different brain areas [16].

Furthermore, the results of this study indicate that the impact of BM on functional connectivity goes beyond local damage. Instead, their influence seems to be global, as FC was disrupted not only in the affected hemisphere, but also in the contralateral hemisphere, and not only in the sensorimotor network, but also in the other networks studied. This is consistent with the conclusions of previous studies on primary brain tumors showing impaired FC even in regions distant from the tumor, such as the contralateral hemisphere [17,18,19,20]. This supraregional impact could be the result of disruption of long-distance partially interhemispheric projections to neuronally connected regions, which impairs the information flow of functionally linked and cooperating areas [20,37]. Similar mechanisms have already been proposed in the context of stroke [38,39]. Aside from that, several patients in this study had already undergone previous therapies. Alterations in fMRI signals [40], as well as FC, might be induced by both chemotherapy [41,42,43] and radiotherapy [44,45]. Moreover, a possible explanation for altered FC in the presence of BM could also be cerebral reorganization [46]. A study using fMRI in patients with brain lesions in eloquent language centers showed neuronal plasticity through greater involvement of the non-dominant hemisphere in language function [47]. Neuronal reorganization has also been shown for motor areas, wherein especially the SMA seems to play a central role [48,49,50]. Additionally, altered FC in brain tumor patients was postulated as the cause of neuropsychological deficits commonly present in patients with brain tumors, including impairment of attention, concentration, and memory, as well as psychomotor slowing, each of which could potentially be explained by the global influence of the tumor [17,18,37]. This could also apply to patients with BM.

This study has several limitations. First, fMRI examinations of the patient and control groups took place in a distinct context using differing paradigms. Patients were examined during preoperative routine clinical diagnostic procedures, whereas controls were tested voluntarily for research purposes. Thus, the influence of possible differences between scanners and examination procedures cannot be determined. Secondly, although the controls were selected to match sex, age matching was not possible. Therefore, age-dependent changes between the patient and control groups cannot be ruled out. Thirdly, metastases were present in both hemispheres in twelve cases, which may account for the fact that PSC values were only in one test significantly different between affected and “unaffected” hemispheres. Lastly, a rather heterogeneous group of patients was analyzed in this study. In general, it is difficult to make a statement about the cause of the neurological deficits of patients and the changes in FC, as the metastasis is not the only possible cause. In addition to the age difference and differences in examination procedures between the two groups, the previous therapies and neuronal plasticity also play a role. An elaborate investigation of these individual aspects in this study would have required a further subdivision of the sample, resulting in a reduction in statistical power and a reduction in the informative value of these analyses, and was therefore not performed.

However, our study has also several strengths. First of all, an elaborated paradigm was applied during the fMRI examination, which allowed for the identification of the eloquent motor cortices in all patients. As the acquisition of the fMRI images took place within the course of routine clinical diagnostics, patients were not exposed to any additional burden. Furthermore, for all patients, histological confirmed diagnosis of brain metastasis was available. By dividing patients according to the mainly affected hemisphere and by always examining the extremities of both sides, a good comparability of the fMRI signal between the affected and contralateral hemisphere was achieved. Lastly, the groups of right and left hemispherically affected patients were tested for a difference in metastatic volume, showing no significant difference.

We therefore conclude that the study and results described here provide important insights into the behavior and influence of BM during fMRI examination.

## 5. Conclusions

Similar to studies in primary brain tumors, we found a qualitative attenuation of the BOLD signal in the metastasis-affected hemisphere compared to the contralateral hemisphere, as well as considerably significant changes in functional connectivity (FC) in the presence of BM in this study. Alterations in FC were not limited to a specific area of the brain, but were also present in regions distant from the lesion. This indicates that metastases cause damage to the integrity of the brain that goes beyond the local damage of brain tissue. This supraregional damage or disturbance of the neuronal networks could explain neuropsychologic deficits in patients suffering from BM, such as impaired attention, concentration, and memory, as well as reduced psychomotor speed found in other studies.

## Figures and Tables

**Figure 1 cancers-16-02010-f001:**
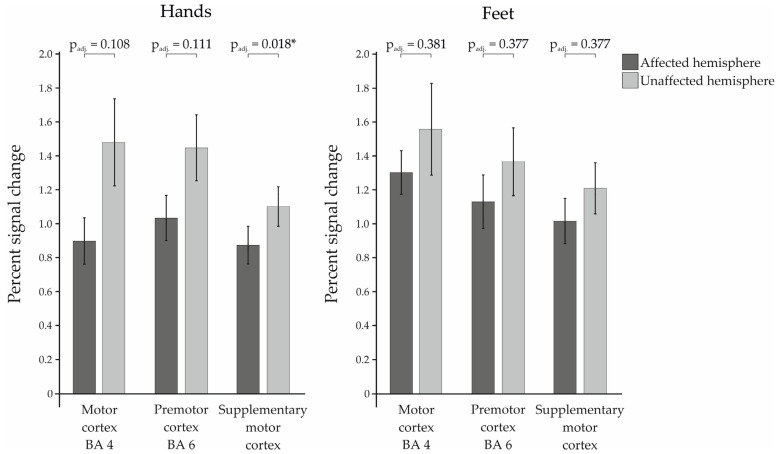
Percent signal changes of the examined Regions of Interest within the patient group for affected vs. the unaffected hemispheres. Statistically significant results are marked with * for *p* < 0.05.

**Figure 2 cancers-16-02010-f002:**
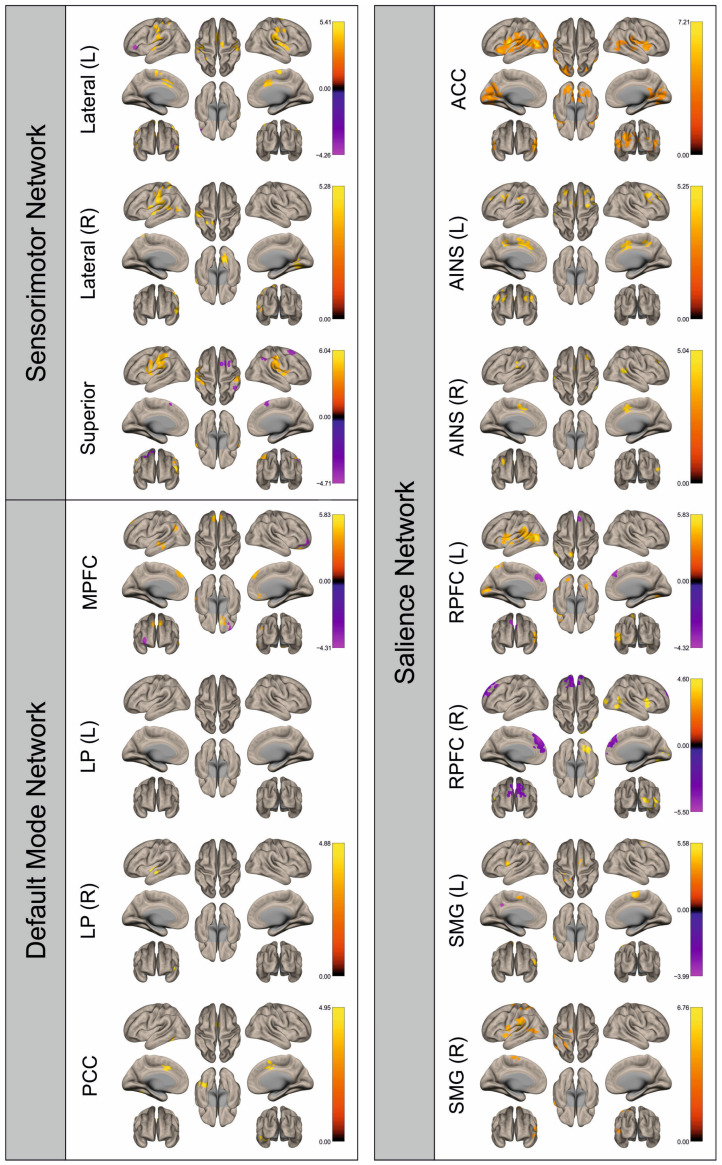
Surface maps showing significant differences in functional connectivity between the patients with a left hemispheric metastasis and their matched controls for the paradigm of the left hand. Areas shown in red to yellow are more strongly connected to the examined seeds in controls, while areas shown in blue to purple are more strongly connected to the seeds in patients. Abbreviations: L: left, R: right, MPFC: medial prefrontal cortex, LP: lateral parietal, PCC: posterior cingulate cortex, ACC: anterior cingulate cortex, AINS: anterior insula, RPFC: rostral prefrontal cortex, SMG: supramarginal gyrus.

**Figure 3 cancers-16-02010-f003:**
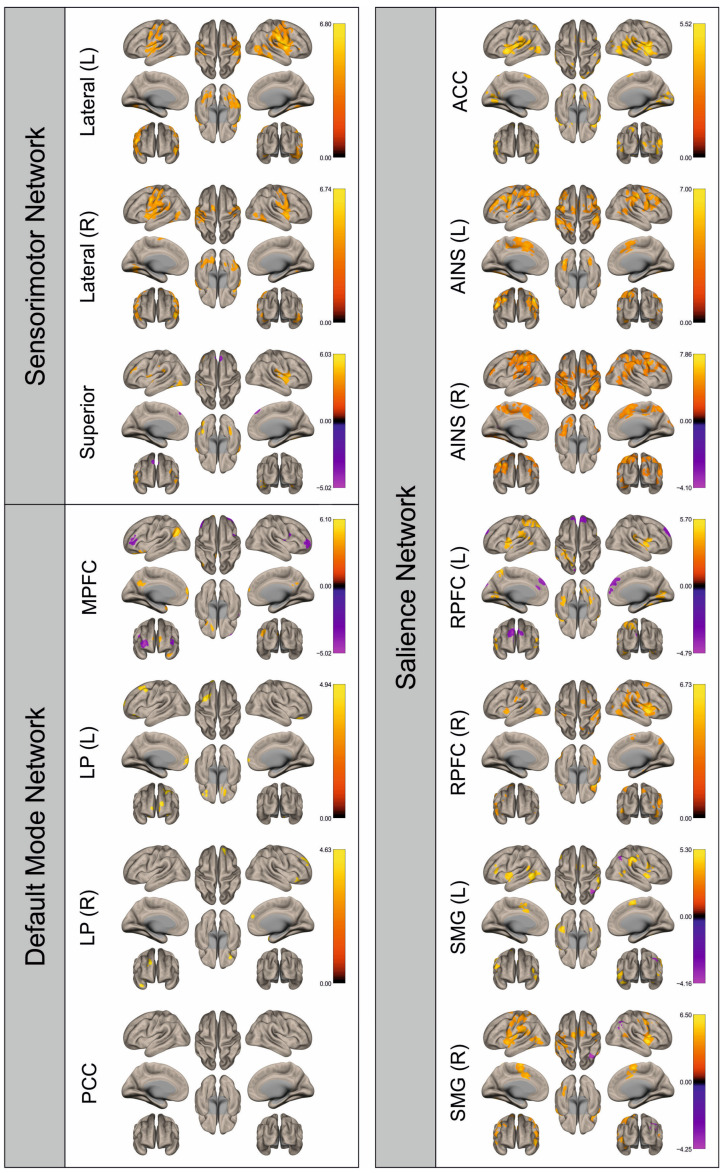
Surface maps showing significant differences in functional connectivity between the patients with a left hemispheric metastasis and their matched controls for the paradigm of the right hand. Areas shown in red to yellow are more strongly connected to the examined seeds in controls, while areas shown in blue to purple are more strongly connected to the seeds in patients. Abbreviations: L: left, R: right, MPFC: medial prefrontal cortex, LP: lateral parietal, PCC: posterior cingulate cortex, ACC: anterior cingulate cortex, AINS: anterior insula, RPFC: rostral prefrontal cortex, SMG: supramarginal gyrus.

**Figure 4 cancers-16-02010-f004:**
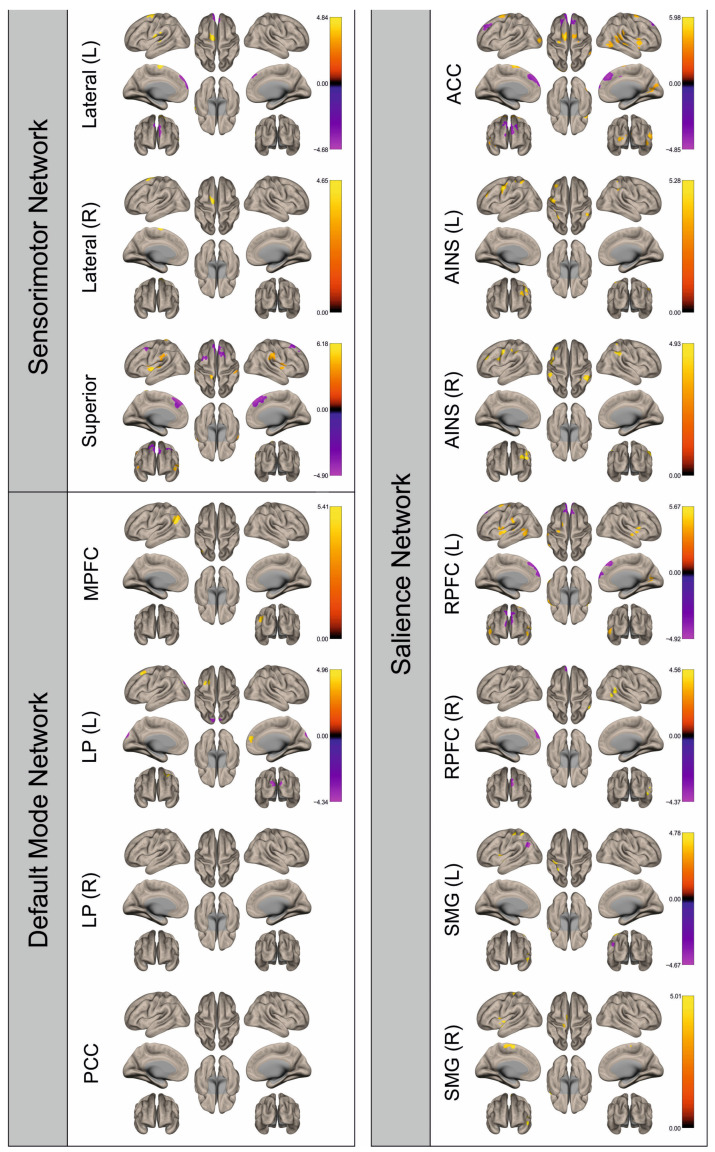
Surface maps showing significant differences in functional connectivity between the patients with a left hemispheric metastasis and their matched controls for the paradigm of the left foot. Areas shown in red to yellow are more strongly connected to the examined seeds in controls, while areas shown in blue to purple are more strongly connected to the seeds in patients. Abbreviations: L: left, R: right, MPFC: medial prefrontal cortex, LP: lateral parietal, PCC: posterior cingulate cortex, ACC: anterior cingulate cortex, AINS: anterior insula, RPFC: rostral prefrontal cortex, SMG: supramarginal gyrus.

**Figure 5 cancers-16-02010-f005:**
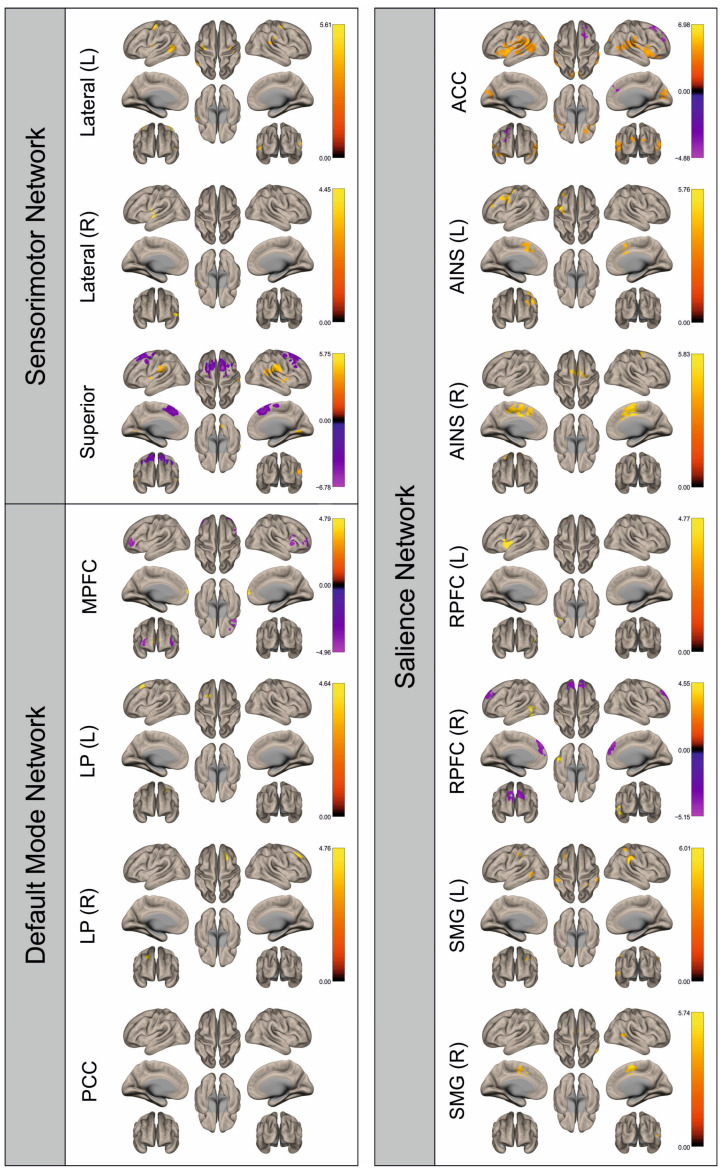
Surface maps showing significant differences in functional connectivity between the patients with a left hemispheric metastasis and their matched controls for the paradigm of the right foot. Areas shown in red to yellow are more strongly connected to the examined seeds in controls, while areas shown in blue to purple are more strongly connected to the seeds in patients. Abbreviations: L: left, R: right, MPFC: medial prefrontal cortex, LP: lateral parietal, PCC: posterior cingulate cortex, ACC: anterior cingulate cortex, AINS: anterior insula, RPFC: rostral prefrontal cortex, SMG: supramarginal gyrus.

**Figure 6 cancers-16-02010-f006:**
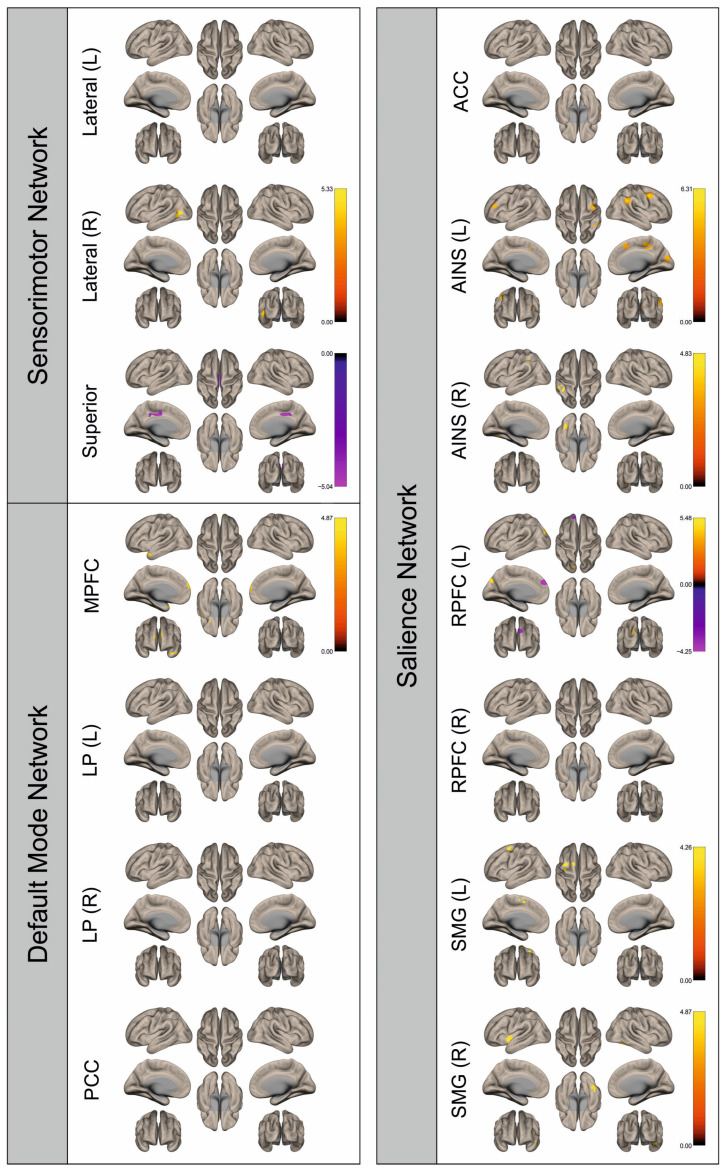
Surface maps showing significant differences in functional connectivity between the patients with a right hemispheric metastasis and their matched controls for the paradigm of the left hand. Areas shown in red to yellow are more strongly connected to the examined seeds in controls, while areas shown in blue to purple are more strongly connected to the seeds in patients. Abbreviations: L: left, R: right, MPFC: medial prefrontal cortex, LP: lateral parietal, PCC: posterior cingulate cortex, ACC: anterior cingulate cortex, AINS: anterior insula, RPFC: rostral prefrontal cortex, SMG: supramarginal gyrus.

**Figure 7 cancers-16-02010-f007:**
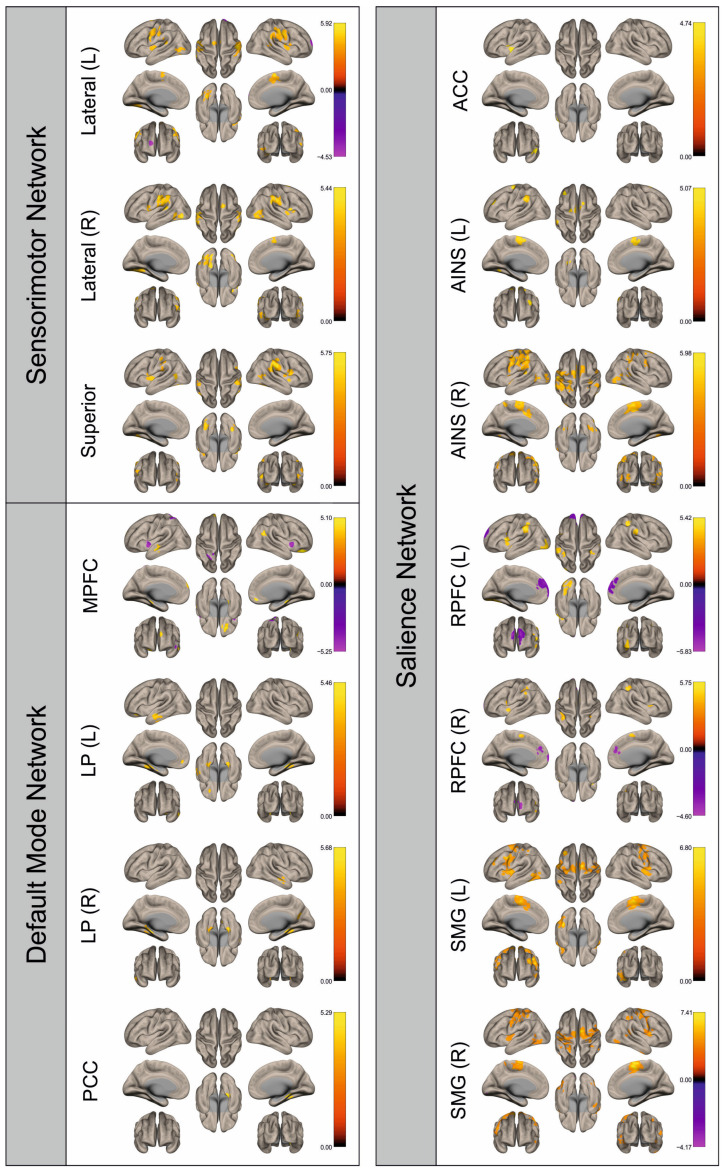
Surface maps showing significant differences in functional connectivity between the patients with a right hemispheric metastasis and their matched controls for the paradigm of the right hand. Areas shown in red to yellow are more strongly connected to the examined seeds in controls, while areas shown in blue to purple are more strongly connected to the seeds in patients. Abbreviations: L: left, R: right, MPFC: medial prefrontal cortex, LP: lateral parietal, PCC: posterior cingulate cortex, ACC: anterior cingulate cortex, AINS: anterior insula, RPFC: rostral prefrontal cortex, SMG: supramarginal gyrus.

**Figure 8 cancers-16-02010-f008:**
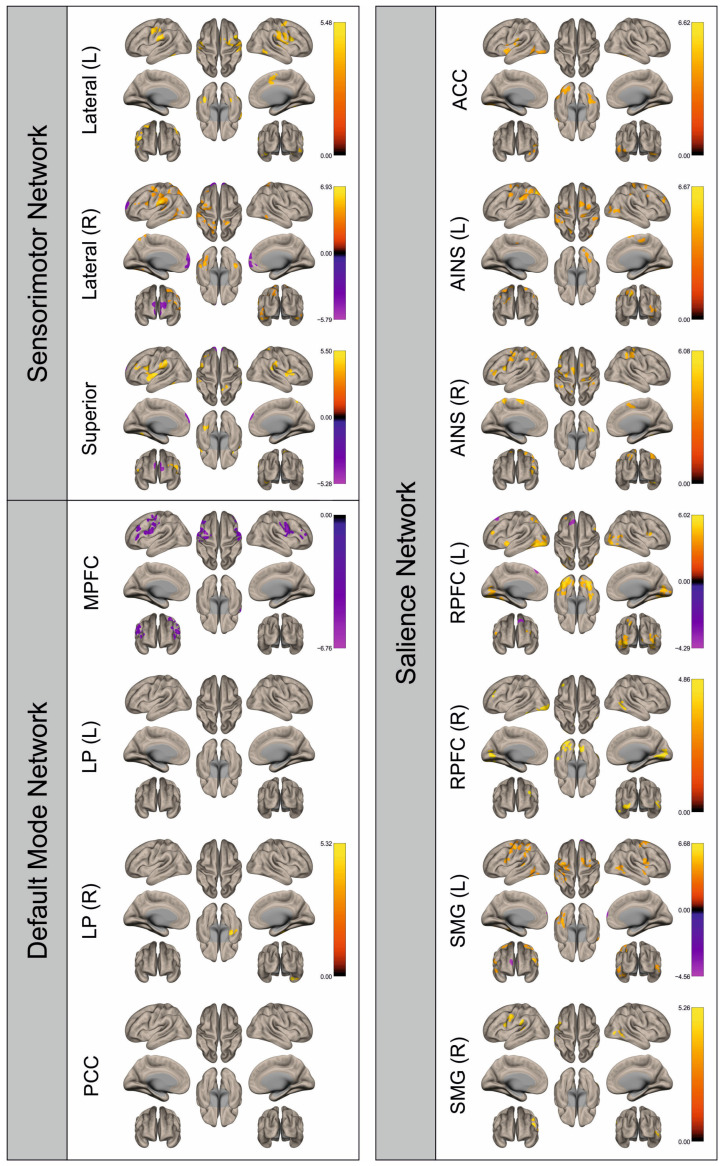
Surface maps showing significant differences in functional connectivity between the patients with a right hemispheric metastasis and their matched controls for the paradigm of the left foot. Areas shown in red to yellow are more strongly connected to the examined seeds in controls, while areas shown in blue to purple are more strongly connected to the seeds in patients. Abbreviations: L: left, R: right, MPFC: medial prefrontal cortex, LP: lateral parietal, PCC: posterior cingulate cortex, ACC: anterior cingulate cortex, AINS: anterior insula, RPFC: rostral prefrontal cortex, SMG: supramarginal gyrus.

**Figure 9 cancers-16-02010-f009:**
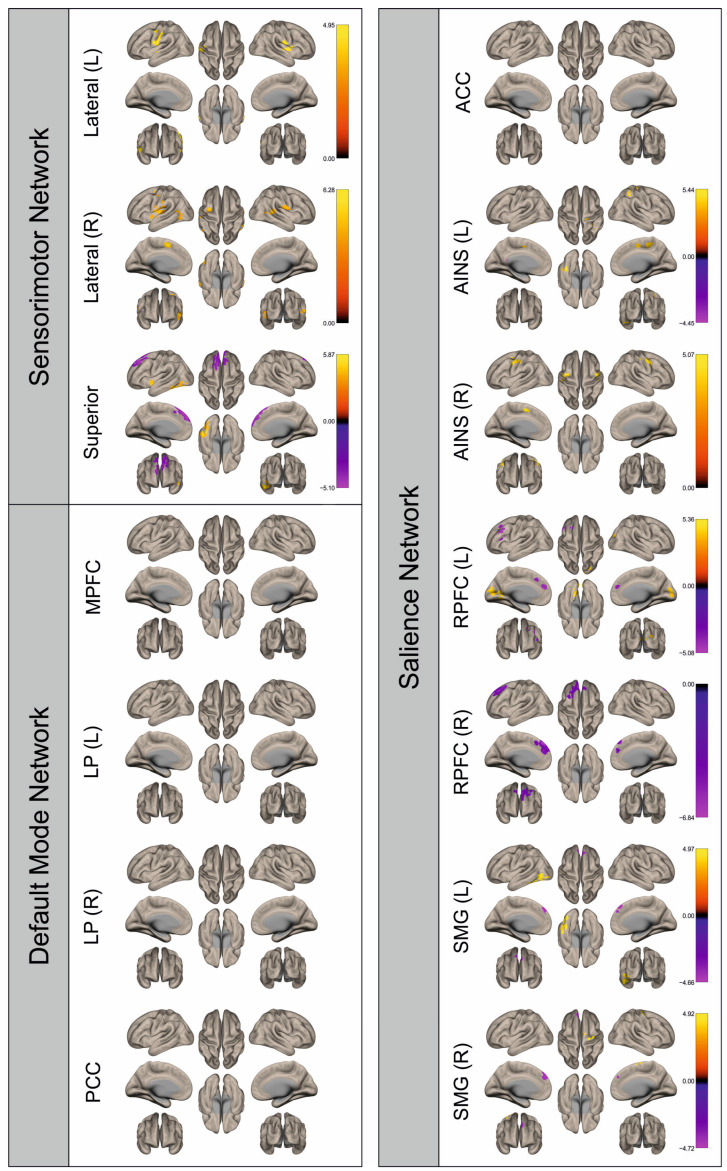
Surface maps showing significant differences in functional connectivity between the patients with a right hemispheric metastasis and their matched controls for the paradigm of the right foot. Areas shown in red to yellow are more strongly connected to the examined seeds in controls, while areas shown in blue to purple are more strongly connected to the seeds in patients. Abbreviations: L: left, R: right, MPFC: medial prefrontal cortex, LP: lateral parietal, PCC: posterior cingulate cortex, ACC: anterior cingulate cortex, AINS: anterior insula, RPFC: rostral prefrontal cortex, SMG: supramarginal gyrus.

**Table 1 cancers-16-02010-t001:** Sample sizes, sociodemographic, and clinical data of the patients.

Paradigm		Total	Left Hemisphere	Right Hemisphere
Hand movement	N	29	16	13
	Male	13	8	5
	Female	16	8	8
	Mean age in years (range)	59.90 (31–85)	58.50 (31–74)	61.62 (36–85)
	Lung carcinoma	11	7	4
	Breast cancer	5	2	3
	Melanoma	4	2	2
	Other primary tumors	7	3	4
	Unknown primary tumor	2	2	
	Systemic therapy + radiotherapy	4	3	1
	Systemic therapy	11	3	8
	Radiotherapy	1		1
	No previous therapy	10	9	1
	Unknown previous therapies	3	1	2
Foot movement	N	16	7	9
	Male	7	3	4
	Female	9	4	5
	Mean age in years (range)	57.63 (36–67)	57.57 (48–67)	57.67 (36–67)
	Lung carcinoma	8	4	4
	Breast cancer	3	1	2
	Melanoma	2	1	1
	Other primary tumors	2		2
	Unknown primary tumor	1	1	
	Systemic therapy + radiotherapy	1		1
	Systemic therapy	7	1	6
	No previous therapy	6	5	1
	Unknown previous therapies	2	1	1

## Data Availability

The data presented in this study are available on request from the corresponding author. The data are not publicly available due to patients’ privacy.

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
