# Peer review of "Attenuation of the BOLD fMRI Signal and Changes in Functional Connectivity Affecting the Whole Brain in Presence of Brain Metastasis"

_cancers, 2024, doi:10.3390/cancers16112010_

Round 1

Reviewer 1 Report

Comments and Suggestions for Authors

I believe that the results regarding diffuse changes in functional connectivity in the presence of brain metastases are novel, interesting in content, and worthy of acceptance, despite the paucity of reports in brain metastases.

However, I think there is room for improvement in the presentation of the results, as some of the figures are difficult to read.

1. Please modify figures of brain showing functional connectivity. Each figure is too small to see so it is hard to tell if it hurts inside or outside.

2. Please include any additional comments on the figures showing functional connectivity. I believe that doing so will emphasize the results I wish to argue for in this research paper and make it easier for the reader to understand.

Comments on the Quality of English Language

Good

Reviewer 2 Report

Comments and Suggestions for Authors

Thank you for giving the opportunity to review the paper entitled: “Attenuation of the BOLD fMRI signal and diffuse changes in functional connectivity affecting the whole brain in presence of brain metastasis”. Although this is an interesting study, it is not clear how to apply these findings in real clinical practice. The data analysis is not very clear, and I have the following comments:

1, Title: “diffuse changes”, do you mean diffusion MRI or something else. In the paper, this is no report for the diffuse changes in the paper.

2, Table 1 in both page 3 and 4. It is better to put the table in one page only.

3, line 122-123, Seventeen patients had various other neurological deficits. This may also lead to connectivity changes in your results, e.g., Figure 2-5. How can you distinguish the FC attenuation is caused by BM, but not due to other neurological deficits?

4, line 202~203, please check. For activation detection, the design matrix includes HRF, not convolve with HRF.  

5,line 211, PSC, how did you calculate the PSC? Did you do: (mean(task block)-mean(rest ))/mean(rest)? Or did you do spatial PSC?  Why there is no negative PSC value in Figure 1?

6, line 218: First and second level seed-to-voxel analyses? Do you mean to define a seed region, and then using the mean value of the seed region and do correlation with the time series from all voxels?

7, line 233, PSC. It seems you calculate PSC in each region (spatially), how did you consider the temporal changes of BOLD signal?

8,line 252~253, difference in FC, how did you measure the FC. From the seed-to-voxel analysis, is the difference of correlation coefficients?

9, figure 2~5, after I zoomed the figure to 800%, I still cannot read the number for the colour bar.

10. Line 291, PSC and FC abbreviations can be used, as these have been mentioned previously.

Comments on the Quality of English Language

English is good. 

Reviewer 3 Report

Comments and Suggestions for Authors

The authors investigated the influence of BM on 32 hemodynamic brain signals deriving from functional magnetic resonance imaging (fMRI) and FC. Generally speaking, the experiments were well designed and the results are interesting. It is suggested to accept this manuscript after minor revisions.

Q1, for figures 2-5, it looks crowded. Please make it clearer to readers.

Q2, please clarify the pros and cons for fMRI in brain metastasis study, as compared to other technologies, like CT, PET, SPET, etc.
